# The Influence of Active and Passive Procrastination on Academic Performance: A Meta-Analysis

Niek Sebastiaan Kooren [1,*] , Christine Van Nooijen [1] and Fred Paas [1,2]

1 Department of Psychology, Education, and Child Studies, Erasmus University Rotterdam, 3062 PA Rotterdam, The Netherlands; vannooijen@essb.eur.nl (C.V.N.); paas@essb.eur.nl (F.P.)
2 School of Education, University of New South Wales, Sydney, NSW 2052, Australia
* Correspondence: 559855nk@eur.nl

**Abstract:** The relationship between academic performance and procrastination has been well documented over the last twenty years. The current research aggregates existing research on this topic. Most of the studies either find no result or a small negative result. However, recent studies suggest that procrastination can have a positive influence on academic performance if the procrastination is active instead of passive. To analyse the effect of active procrastination on academic performance, a meta-analysis was conducted. The analysis includes 96 articles with 176 coefficients including a combined average of 55,477 participants related to the correlation between academic performance and procrastination. The analysis uncovered a modest negative correlation between academic performance and procrastination overall. Importantly, the type of procrastination exerted a substantial impact on the strength of this correlation: active procrastination demonstrated a small positive effect size, whereas passive procrastination registered a small negative effect size. Additionally, participant-specific characteristics and indicators further modulated the magnitude of the correlation. The implications of this research extend to underscoring a potential beneficial aspect of procrastination, specifically elucidating how certain types of procrastination can positively influence academic performance.

**Keywords:** procrastination; academic performance; meta-analysis; active procrastination; passive procrastination





## 1. Introduction

While academic performance is relatively straightforward to define as the extent to which someone has achieved their academic goals, such as sufficient grades and diplomas [1], the definition of procrastination proves more complex due to its multifaceted nature and various types. A common method employed to classify these types of procrastination is to distinguish between trait (or general) procrastination and state (or situational) procrastination. Trait procrastination refers to a general tendency to procrastinate, while situational procrastination is triggered by a specific context [2]. One of the most pressing forms of situational procrastination occurs in academic contexts. Academic procrastination is defined as the delay of completion or initiation of academic tasks to the point where the delay can be described as irrational [3]. A study conducted by Sirin [4] revealed a moderate correlation between situational and academic procrastination. Within academic procrastination, one can distinguish between behavioural or avoidant procrastination, defined as the postponement of action, and decisional procrastination, defined as the postponement of decisions [5].

Over the past four decades, research has consistently indicated a high prevalence of procrastination. An early study reported that 46% of students exhibited signs of academic procrastination [6]. Another study by Harriot and Ferrari [7] reported that chronic procrastination affects 20% of adults. A recent study by Popova and Pronenko [8], based on

self-reported data, indicated that up to 60% of college students exhibit significant levels of academic procrastination. So, the prevalence rates of procrastination vary considerably across studies and specific types of procrastination.

Unfortunately, academic procrastination can come at a serious cost to well-being, self-efficacy, and academic performance, while increasing test anxiety and academic stress [9]. Other research has linked procrastination to anxiety, lack of self-esteem, and depression [10]. Multiple meta-analyses have consistently found a negative correlation between procrastination and academic performance, with effects ranging from minor to moderate in severity [11–16].

The decline in academic performance associated with procrastination can be attributed to both direct causes and a multitude of indirect effects. A direct influence of procrastination is the delay of academic tasks, reducing the time available for studying or project work, which in turn diminishes performance due to time constraints. This behaviour may hinder the utilisation of distributed practice, a technique known to enhance memory retention when applied effectively [17]. This typically leads to cramming, often compromising sleep quality [18]. Indirectly, procrastination may influence academic performance through various factors including lack of motivation, suboptimal learning approaches [19], increased anxiety, aversiveness, loss of productivity, sense of social disapproval, stress [20], and poor self-regulation [21]. The many indirect effects complicate the assessment of procrastination's direct impact on academic performance. Moreover, most studies report correlational data, which do not clarify the causal relationships between these variables. It is possible that there is no direct causal link, and that the observed low academic performance and procrastination are both outcomes of other underlying factors. This complexity contributes to ongoing debates regarding the relationship between procrastination and academic performance [22].

While the existing literature has predominantly concentrated on the negative aspects of procrastination, some types may lead to benefits. Chu and Choi [23] argue for a differentiation between passive and active procrastination. Passive procrastination is characterised by indecision to act, whereas active procrastination is marked by a preference and deliberate decision to work under pressure. Alternative terminologies for these procrastination types have been proposed, such as structured and unstructured procrastination [24], or strategic delay and procrastination [25], serving similar conceptual roles as active and passive procrastination, respectively. According to Kim et al. [26], active and passive procrastination differ in their correlation with personality and academic performance. They report that passive procrastination is negatively related to the ability to meet deadlines, extraversion, agreeableness, conscientiousness, and Grade Point Average (GPA), but positively related to neuroticism, which is the inclination to experience adverse emotions. Conversely, active procrastination has positive low to high correlations with a preference for pressure, ability to meet deadlines, outcome satisfaction, extraversion, and a negative correlation with neuroticism. Although active procrastination has no significant correlation with GPA, it is preferable relative to passive procrastination, which has a medium negative correlation with GPA.

Knowing this distinction between active and passive procrastination, one can divide the different procrastination measurement tools into those that measure active and passive procrastination (see Appendix A for a comprehensive summary of different procrastination indicators). The summary was compiled by applying the snowball technique to existing literature [27]. This means that an initial pool of articles was searched using the keyword "procrastination assessment", from which each reference list was examined to find further references, and this repeated until no new procrastination indicators were found. The summary in Appendix A lists the most common procrastination indicator tools, including how they assess both procrastination itself and procrastination type. The list does not include workplace-related procrastination, but mostly general, behavioural, avoidant, decisional, and academic procrastination.

Of all the different measurement tools, the Procrastination Assessment Scale for Students (PASS) [6] is the most known and established assessment tool. The PASS assesses procrastination using 12 self-reported items using two subscales related to the frequency and problem of procrastination. Similarly, most other procrastination indicators use self-assessment questionnaires, with the only exception being the Procrastination Checklist Study tasks (PCS) [28], which offer a measurement based on study-related behaviours including time of initiation intention and time of completion. To our knowledge, the only procrastination indicators that look at both passive and active procrastination are the Academic Procrastination Scale (APSM) [24] and the Metacognitive Beliefs about Procrastination scale (MaP) [29]. The APSM assesses both active and passive procrastination by assessing the common characteristics of both types of procrastination in a self-assessment questionnaire. The MaP assesses metacognitive beliefs about procrastination using 16 self-assessment items that result in an indication of passive, active, or non-procrastinator. There is one indicator that looks at active procrastination only, named the Active Procrastination Scale (APSCM) [30]. The APSCM is a 16-item self-assessment questionnaire that assesses four characteristics of procrastination: satisfaction with an outcome, ability to meet deadlines, intention to procrastinate, and preference for time pressure. The rest of the procrastination indicators focus on the negative aspects of procrastination, and will therefore be classified as measurements for passive procrastination.

When reviewing works on procrastination in relationship with academic performance, the following performance indicators are prevalent: Grade Point Average (GPA), mid-term and final examination score, assignment grade, quiz, course grade, homework, and American College Test (ACT) [12].

This study will focus on the relationship between procrastination and academic performance. The relationship will be assessed using the research questions: (a) What is the relationship between academic performance and procrastination? (b) Does the type of procrastination, active or passive, influence the relationship with academic performance? (c) Is the relation between academic performance and procrastination influenced by the variables age, type of measurement, academic performance indicators, and procrastination indicators?

In this study, we hypothesise a negative correlation between procrastination and academic performance, consistent with the extant literature [9,11–16,31]. Building on Chu and Choi's [23] framework, we anticipate distinct relationships between types of procrastination and academic performance: a negative correlation for passive procrastination and a positive correlation for active procrastination. We expect that younger students will exhibit stronger correlations between academic performance and procrastination than older students, aligning with the findings of Kim and Seo [12]. With respect to measurement, we expect self-reported data to yield smaller correlations due to inherent variability and reliability issues. For academic performance indicators, we anticipate uniform correlations, contrasting the varied correlations expected from different forms and types of procrastination measures.

This study aims to bring clarity to the mixed findings concerning the relationship between procrastination and academic performance. Various studies have depicted this relationship in contrasting lights: as negative [9], neutral [32], and even positive [23]. With our results we intend to provide more definitive insights into this complex association. A meta-analysis by Kim and Seo [12] found a negative correlation between academic performance and procrastination. However, they found a high variation in the correlation sizes and between different subgroups, such as geographical region, self-reported data, performance indicators, procrastination indicators, and age. Using the newly published data, this study contributes to the scientific knowledge base by extending this research with more published data. This study's findings will equip educators with the critical understanding of when procrastination might negatively or positively influence academic performance.

The research questions will be addressed by synthesising results from previous studies on the topic of procrastination and academic performance. The method used in this study is a meta-analysis, which is used to provide an overview of already published articles in a structured fashion [33] (pp. 3–13).

## 2. Method

Data were sourced from the Web of Science database and accessed through various platforms: Erasmus University Rotterdam Library (EUR Library), free databases via Google Scholar, and Unpaywall [34]. The sample was compiled using a targeted search query incorporating the keywords "procrastination" and "academic performance". The keyword search query included synonyms and terms identified in prior meta-analyses [11–16]. The exact keyword search query stated the following: "procrastination" (Topic) AND "academic performance" OR "performance" OR "accomplishment" OR "academic achievement" OR "GPA" OR "grade" OR "school performance" OR "examination score" OR "ACT" OR "academic success" OR "missing deadline" OR "task completion time" OR "task delay" OR "task preparation time" OR "grade point average" (Topic) and Proceeding Paper or Article (Document Types).

Data research was limited to the year 1975, the earliest year of publication Web of Science can access, until the 13th of May 2023, the date of data extraction. Studies included in the analysis had to meet the criteria listed in Table 1.

**Table 1.** Inclusion and exclusion criteria.

| Number | Inclusion Criteria | Exclusion Criteria |
|:---:|:---:|:---:|
| 1 | The study is empirical in nature. | The study is a review. |
| 2 | The article is on the topic of procrastination and academic procrastination. | The article is not on the topic of procrastination and academic procrastination. |
| 3 | The article is accessible through open databases or via EUR Library. | The article is not accessible through open databases or via EUR Library. |
| 4 | Reported Pearson correlation coefficients between academic performance and procrastination, along with sample sizes. | The article did not report the correct correlation coefficient as per inclusion |
| 5 | Articles are available in English. | Articles are not available in English. |
| 6 | The sample size of the article was less than 10,000 | The sample size of the article was over 10,000. |
| 7 | The sample represents a population with normal learning capabilities. | The study has a sample size consisting of only academically challenged participants. |
| 8 | The study was not retracted. | The study was retracted. |
| 9 | The studies are unique in individual participants, procrastination measures, or academic performance measures. | The studies are not unique in individual participants, procrastination measures, or academic performance measures (copy of article excluded). |
| 10 | The study does not attempt to influence the procrastination of the participants. | The study is on procrastination interventions. |

The studies found through the keyword search were first scanned using the title and abstract to assess their relevance to the topic of procrastination and academic performance. Studies that did not mention the topic of procrastination and academic performance in the title or abstract were excluded from further analysis. Then the studies were scanned with a focus on the inclusion and exclusion criteria, particularly in the Sections 2 and 3. If a study met all criteria, the entire article was assessed, and data were extracted. The following data were extracted: the author(s) name(s), publication year, sample size (N), sample age range (pre-primary, primary school, secondary education, college, adults), geographic location, procrastination indicator (see Appendix A), academic achievement indicator, correlation coefficient(s), type of procrastination (active or passive procrastination), and

type of measurement (self-assessment or external assessment). If a study had altered an existing measure of procrastination and academic performance to an extent where the measure became unique or the study had a unique measurement tool, then the indicator was noted as research specific.

The retrieved studies and excluded studies were visualised using the Preferred Reporting Items for Systematic Reviews and Meta-Analyses (PRISMA) 2020 flow [35] and can be found in Figure 1. The PRISMA flow diagram was used to note the records that were identified, removed, screened, excluded, retrieved, not retrieved, assessed, excluded after assessment, and included.

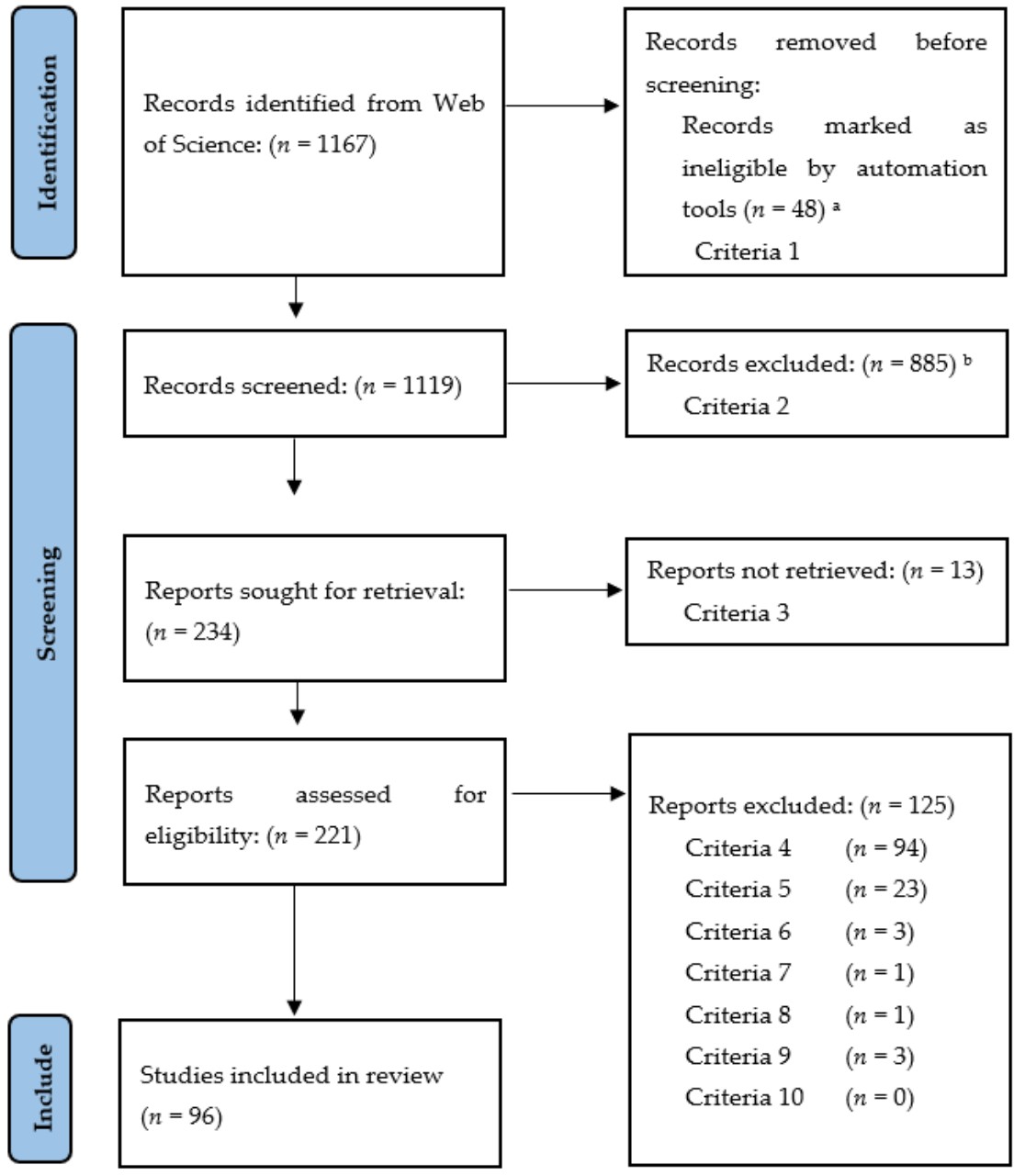

**Figure 1.** PRISMA flow diagram including the number of articles per meta-analytic phase identification, screening, and included. Note. This flow diagram denotes the specific number of articles in every step of the meta-analysis process and how the number of articles was reduced to just those included into the meta-analytic review. [a] Recognized by Web of Science as review articles and excluded by exclusion criteria two. [b] Records are excluded from eligibility assessment as the research topic is irrelevant to the topic of the current research.

When an article reported multiple coefficients, all relevant coefficients were extracted. If an article reported multiple similarly tested procrastination scores over a period of time, then only the score closest to the time that the academic performance score was obtained was noted. If an article used multiple procrastination indicators and academic performance indicators, all variations were extracted. However, for analysis, these results were averaged and weighted to ensure that each study contributed proportionally to its sample size. Coefficients derived from the same participant pool were weighted based on sample size and the total number of relevant coefficients using the formula: ((sample size/number of relevant coefficients)/total number of participants in the meta-analysis) xPearson correlation coefficients. In cases where studies reported correlation coefficients for differing sample sizes drawn from the same participant pool, an average sample size for each study was calculated to determine the total number of participants.

The extracted data were organised using Excel version 2304 and analysed using IBM SPSS Statistics 29. The data were weighted according to the sample size of the individual study. The correlation coefficients *r* were transformed into a *z* statistic using Fisher's *z* transformation, before an average was calculated to make the average *r* less biased [36] when transformed back from an average *z* to an average *r* [37]. Fisher's *z* was also used to calculate significant differences between effect sizes. Effect sizes were characterised as small ($r < 0.2$), small to medium ($r = 0.2$–$0.5$), medium to large ($r = 0.5$–$0.8$), and large ($r > 0.8$) [38]. *p* values would indicate decisive evidence ($p < 0.001$), substantive evidence ($p = 0.001$–$0.01$), positive evidence ($p = 0.01$–$0.05$), or no evidence ($p > 0.05$) [38].

A meta-analysis was planned using either a random-effects model or fixed-effects model, pending the results of the heterogeneity tests. Given that the included studies employed diverse indicators to measure various facets of procrastination, it was anticipated that a random-effects model would be more appropriate. The studies also varied in terms of geographic location, year of publication, time between measurements, sample age, type of procrastination, and performance indicators. These variations likely violate the assumption of identical empirical settings needed to perform a fixed-effects model [39]. A random-effects model approach is in line with most meta-analytic approaches [40]. The estimator used for the meta-analysis was the Restricted Maximum Likelihood (REML) [41].

Apart from the weighted *r*, all test statistics were gathered using the meta-analysis function included in the IBM SPSS 29th edition. The meta-analysis on continuous effect size reported an effect size estimate, homogeneity test statistics, heterogeneity test statistics, Egger's regression-based test, Trim-and-Fill analysis [42], and a Forrest plot. Using the average *z*, a 95 per cent confidence interval [43] was computed separately for the extraction variables where applicable. The average *z* was also used to compute a *p*-value to examine the statistical significance of the correlation coefficients. To perform the weighted *r* analysis, a standard error was calculated using the formula $(1 - r^2)/\sqrt{N - 3}$ to minimise potential bias. This formula was, compared to other formulas, one of the most unbiased methods available to calculate the standard error, only becoming slightly negatively biased for correlations larger than $r = 0.60$ [44].

The data were analysed on homogeneity using the homogeneity statistic *Q* and the heterogeneity statistic $I^2$. The *Q* statistic was used to assess the presence of heterogeneity [45]. This meant that a significant *Q* statistic would result in the use of a random-effects model, as the data were not homogenous, and a non-significant *Q* statistic would result in a fixed-effects model, as the data were homogenous [46]. The $I^2$ was interpreted to quantify the extent of heterogeneity, where $I^2 \approx 25\%$ was defined as low heterogeneity, $I^2 \approx 50\%$ was defined as medium heterogeneity, and $I^2 \approx 75\%$ was defined as high heterogeneity [47].

The potential for publication bias was analysed using the Trim-and-Fill method [48]. A small study bias was assessed using Egger's regression test [49]. These tests are most accurate in detecting publication bias and small study bias when compared to the Tang's regression test and Begg's regression test [50].

The data were analysed for outliers using the interquartile range technique. This technique entails that an outlier must be at least 1.5 times the length of the interquartile

range away from the mean to be identified as a mild outlier. If the outlier is three lengths of the interquartile range away from the mean it was classified as an extreme outlier [51]. However, based on the research by Hansen et al. [39], outliers were not removed due to the variation in procrastination indicators, performance indicators, procrastination facets, and performance facets, which likely resulted in very different effect sizes. This variation in effect sizes was desired for this study as we were interested in overall effect size.

## 3. Results

The search query on procrastination and academic performance resulted in 1167 articles. From those articles, 48 articles were removed as they were identified as reviews by Web of Science. This resulted in 1119 records that were screened based on the title, keywords, and abstract. This screening process deemed 885 articles irrelevant to the current research and excluded them from the full article assessment. This led to the identification of 234 articles, which were subsequently accessed using open databases and resources available through EUR Library. All but 13 articles were successfully retrieved for full assessment. The full assessment identified 96 articles that fit the inclusion and exclusion criteria and basic guidelines. This meant that 125 articles did not meet these requirements. A full rundown of the article selection process is available in Figure 1 and the list of included studies (*n*) is available in Appendix B Table A2.

Articles included in the analysis (*n* = 96) reported a combined number of coefficients (*k* = 176). The articles had a combined average number of participants (*N* = 55,477) and a unique pool of participants ($N_{un}$ = 55,555) based on the unique number of participants per study, excluding the study by Seo [52], as this study used the same participant data as in a study by Seo [18]. The articles show a positive trend towards the year of publication, with more recent years having more published articles.

Before examining the correlation sizes, the heterogeneity was assessed using the *Q* statistic (*Q* = 5354.761, *p* < 0.001) and resulted in a significant statistic, meaning a random-effects model is most appropriate for our data analysis. The high $I^2$ statistic ($I^2$ = 95.7%) suggests the data are highly heterogenous, confirming this choice.

The Trim-and-Fill analysis for publication bias resulted in zero imputed and this suggests that the data do not suffer from publication bias. The Egger's regression test resulted in a significantly low negative intercept (*a* = −0.225, *t* = −6.049, *p* < 0.001), which indicates that studies with smaller sample sizes show larger effects in our data. A visual inspection confirms this test result with hardly noticeable asymmetry in the data, as shown in the funnel plot in Appendix C Figure A1.

However, what is noticeable is the asymmetry in the standard error, with most studies having a standard error smaller than 0.10 and only one study having a standard error larger than 0.20.

The interquartile range technique identified four mild outliers, two positives (*r* = 0.42, *r* = 0.34) and two negatives (*r* = −0.87, *r* = −0.71). These outliers were confirmed by visually inspecting the Forest plot. Together, these outliers are responsible for the sum of 1247 participants, which is about 2.2% of the overall sample size. The outliers have a slightly negative influence on the overall correlation coefficients, with an averaged correlation slightly lower than the overall correlation (*r* = −0.21). These outliers are identified for discussion purposes but not removed due to the allowed variance in methods.

The weighted mean for the correlation between procrastination and academic performance was significant (*r* = −0.18). The weighted means of the correlations for the type of procrastination were all significant, including active (*r* = 0.15), passive (*r* = −0.17), and not provided (*r* = −0.43). All mentioned indicators for academic performance were significant for weighted *r*: average grade (*r* = −0.28), course grade (*r* = −0.17), exam (*r* = −0.22), and GPA (*r* = −0.20). The listed indicator API (*r* = −0.03) had a non-significant weighted *r*, while the rest were significant: APSCM (*r* = 0.16), dilatory behaviour (*r* = −0.24), GPS (*r* = −0.22), PASS (*r* = −0.17), and TPS (*r* = −0.22). The sample age ranges resulted in correlations that were marginally significant and varied across groups, including adults

($r = -0.39$), college ($r = -0.22$), and secondary education ($r = -0.11$). The type of reported academic performance resulted in two significant weighted correlations for externally gathered ($r = -0.18$) and self-reported ($r = -0.22$) performance, but an insignificant result when performance was not provided ($r = 0.00$). The type of reported procrastination resulted in two significant weighted correlations for externally gathered ($r = -0.49$) and self-reported ($r = -0.16$) procrastination data.

The unweighted correlations (estimated $r$) and relevant statistics for all collected variables can be viewed in Table 2 and Appendix D Tables A3 and A4.

**Table 2.** The correlations between academic performance and procrastination and other variables.

| Total | $k$ | Sample Average $N$ | Weighted $r$ | Estimated $r$ | Confidence Interval (95%) | $Q-$Value | $I^2$ (%) |
|---|---|---|---|---|---|---|---|
| Procrastination– performance | 176 | 55,477 | −0.18 | −0.19 | [−0.223, −0.163] | 5354.761 | 95.7 |
| *Type of Procrastination and Performance* | | | | | | | |
| Active | 14 | 2377 | 0.15 | 0.13 * | [0.035, 0.266] | 125.450 | 89.3 |
| Passive | 137 | 53,398 | −0.17 | −0.20 | [−0.230, −0.174] | 3160.315 | 94.0 |
| Not provided | 25 | 2868 | −0.43 | −0.33 | [0.411, −0.245] | 537.908 | 94.5 |
| *Indicator of Performance* | | | | | | | |
| Average grade | 23 | 7252 | −0.28 | −0.27 | [−0.332, −0.217] | 134.398 | 86.5 |
| Course grade | 25 | 14,915.5 | −0.17 | −0.28 | [−0.396, −0.168] | 2927.722 | 98.9 |
| Exam | 39 | 7176 | −0.22 | −0.16 | [−0.220, −0.106] | 554.141 | 91.7 |
| GPA | 60 | 16,743.5 | −0.20 | −0.18 | [−0.224, −0.138] | 589.932 | 91.0 |
| *Indicator of Procrastination* | | | | | | | |
| API | 13 | 2999 | −0.34 | −0.27 | [−0.343, −0.189] | 87.730 | 87.0 |
| APSCM | 22 | 2377 | 0.16 | 0.07 ** | [0.003, 0.143] | 169.589 | 87.5 |
| Dilatory behaviour | 10 | 772 | −0.23 | −0.24 | [−0.297, −0.181] | 19.377 ** | 49.9 |
| GPS | 11 | 2993 | −0.22 | −0.31 | [−0.431, −0.190] | 81.074 | 90.4 |
| PASS | 21 | 5424 | −0.17 | −0.14 | [−0.217, −0.068] | 335.113 | 94.5 |
| TPS | 20 | 8479 | −0.22 | −0.24 | [−0.286, −0.190] | 109.431 | 82.5 |
| *Sample Age Range* | | | | | | | |
| Adults | 1 | 83 | −0.39 | −0.39 | [−0.576, −0.204] | SR | SR |
| College | 158 | 33,048.5 | −0.22 | −0.19 | [−0.221, −0.155] | 4380.464 | 94.7 |
| Secondary education | 17 | 22,345.5 | −0.11 | −0.23 | [−0.297, −0.153] | 624.412 | 97.4 |
| *Reported Performance* | | | | | | | |
| Externally gathered | 98 | 24,864 | −0.18 | −0.21 | [−0.259, −0.169] | 3813.492 | 96.2 |
| Self−reported | 76 | 25,190 | −0.22 | −0.17 | [−0.206, −0.163] | 1124.649 | 93.6 |
| Not provided | 2 | 6540 | 0.00 *** | −0.14 *** | [−0.439, 0.162] | 27.871 | 96.4 |
| *Reported Procrastination* | | | | | | | |
| Externally gathered | 24 | 2256 | −0.49 | −0.38 | [−0.467, −0.289] | 720.136 | 96.0 |
| Self−reported | 152 | 53,676 | −0.16 | −0.16 | [−0.194, −0.135] | 2329.641 | 94.7 |

Note. Variables weighted $r$, estimated $r$, and $Q$-value are significant at $p < 0.001$ unless marked with an asterisk. SR = statistics cannot be computed because this subgroup contains a single record. The abbreviations found in the 'Indicator of Procrastination' section are explained in Appendix A. * $p < 0.01$, ** $p < 0.05$, *** $p > 0.05$.

## 4. Discussion

The present study was undertaken to aggregate existing research on the relationship between procrastination and academic performance. Our findings suggest a small negative correlation between the two variables. This could imply that those who procrastinate may either have lower academic aptitude, underperform due to procrastination, or develop procrastination tendencies due to poor academic performance. Notably, the data exhibited significant heterogeneity, indicating that the included studies may have focused on disparate aspects or types of procrastination. This was partially confirmed by analysing the most popular procrastination measures (see Appendix A, which mentions a variety of measurement techniques, aspects of procrastination, and types of procrastination that likely contribute to the observed heterogeneity). However, additional factors such as participant

characteristics, geographical location, performance metrics, data collection methods, and temporal aspects of the study further contributed to the observed heterogeneity.

This result is very similar to an earlier meta-analysis by Kim and Seo [12], but their study reported a slightly lower effect size. The result is also comparable to those of van Eerde [16] and Steel [15], with some comparable effect sizes, some slightly larger and some slightly smaller. However, the result did differ significantly from the average effect size reported by Setayeshi Azhari [14], which reported a small to medium overall effect size compared to the small effect size found in this study. Overall, our results mostly align with those of previous studies.

Of interest were the results related to the type of procrastination. Active procrastination showed a small positive relationship with academic performance, while passive procrastination had a small negative relationship with academic performance. Noticeable is that the studies wherein the type of procrastination was not specified showed a small to medium negative correlation with academic performance. The results of the type of procrastination closely matched the accompanying procrastination indicators, as most passive indicators reported similar correlations to the correlation between passive procrastination and academic performance. Active procrastination and academic performance showed similar effect sizes compared to APSCM and academic performance. The externally gathered procrastination data aligned closely with the non-specified type of procrastination. The effect sizes of the externally gathered procrastination data were significantly higher compared to previous meta-analyses on this type of data [12,53].

Another interesting variable is the sample age range, as adults appear to have the highest effect size for the relationship between procrastination and academic performance, followed by college students with about half the effect size, and then followed by students in secondary education with half the effect size of the college students. This being said, only one study in this dataset used adults as their participant population, so this phenomenon is not strongly supported. It is, however, interesting to consider, as this finding directly opposes the results found by Kim and Seo [12].

Due to the correlational nature of the extracted coefficients, a causal relationship cannot be confirmed. It could be the case that procrastination causes poor academic performance, or that people with low academic performance tend to procrastinate more compared to those with high academic performance. These relationships may also exist simultaneously.

These results can be used to answer the research questions. The first research question on the relationship between academic performance and procrastination can be answered by the overall correlation statistic, stating that there is a bivariate small negative correlation between academic performance and procrastination. This is in line with the hypothesis and previous research. The second research question was on the influence of the type of procrastination on the correlation with academic performance. Passive procrastination was found to have a small negative correlation with academic performance, and active procrastination had a small positive correlation with academic performance. These results confirm the hypothesis that active procrastination is positively correlated with academic performance and passive procrastination is negatively correlated with academic performance. Contrary to our hypothesis, the impact of age on effect sizes was inversely related, leading us to reject this aspect of our hypothesis. As for the influence of measurement type, our findings partially corroborate and contradict our initial expectations. Specifically, self-reported data for academic performance yielded larger effect sizes compared to externally collected data. Conversely, for procrastination, self-reported data produced smaller effect sizes relative to externally sourced data. The results of the performance indicators were in line with the hypothesis, with all indicators having the same directionality in correlation, all hovering around a small to medium correlation. The hypothesis related to the procrastination indicators was also confirmed, as procrastination indicators appear to correlate with the type of procrastination that they are intended to measure.

This study has several limitations. One such factor is the possibility of publication bias. Even though the Trim-and-Fill analysis concluded there was no publication bias, the

test is known to perform poorly in the presence of high heterogeneity [48,54], which is the case for this dataset. The second bias test, Egger's regression test, resulted in a negative small study bias, meaning that small studies show more extreme effect sizes compared to larger studies. This could be negligible for a number of reasons: most studies in the dataset have a relatively high sample size, the weighted *r* makes these studies less impactful, and the reported bias was relatively small. Knowing this does not rule out the possibility of publication bias.

The weighted *r* also has limitations. Using a weighted correlational approach means that large studies have more of an impact on the overall correlation statistic. However, one could argue that it is not fair to assign more weight to large studies, as they might have less controlled methods compared to smaller studies, where the study process can be more easily controlled. This method makes the overall correlation coefficient more biased towards the facets of procrastination and academic performance that are measured by larger studies. To combat this limitation, both the weighted *r* and the unweighted estimated *r* have been reported for comparison. The difference between the weighted *r* and the unweighted *r* was very small for most variables.

Self-reported data can influence the objectivity of the correlation. Self-reported data on academic performance such as GPA generally correlate well with objective GPA, but the correlation can vary depending on many of the participants' characteristics [55]. The objective measures of procrastination might also be questionable as they generally tend to focus on measures such as delay, absence, and time, which do not cover the full extent of procrastination and cannot distinguish between active and passive delay related to the types of procrastination [56].

Articles used in this study span a wide timeframe in which the academic setting has changed. Particularly during the years 2020 to 2022, online education became more prevalent due to the pandemic [57]. This could have influenced the relationship between procrastination and academic performance. It could be that the correlation between the variables has decreased as the gap in academic performance has declined due to COVID-19. It could also have increased the correlation, as COVID-19 could have impacted procrastinators more than others in terms of academic performance.

Results on active procrastination and the comparison with passive procrastination should be interpreted more critically compared to the rest of the results. The definitions of procrastination and active procrastination differ substantially, as one is considered irrational and the other deliberate and intentional. A review study by Klingsieck [25] compared popular definitions of procrastination and strategic delay, which is similar to active procrastination, and found that strategic delay differs in facets such as irrationality, awareness, and discomfort. It might not be fair to compare active and passive procrastination as they analyse different facets. This difference is visible in the effects sizes, as they differ significantly. This difference in definition might limit the comparison strength between the two procrastination types.

The review process of this study also has some limitations. Due to the method of screening, if the appropriate variables were not mentioned in the title, abstract, or keywords, the study was not included in the full article review. This method allowed us to find many articles, but may have caused some relevant publications to go unnoticed.

The selection and screening of articles were done by the first author using one database, and are therefore prone to more mistakes in data assessment and extraction compared to the use of multiple assessors. Potentially overlooked articles could have provided different or more definitive results. The results of this study could be biased towards the journals that are supported by Web of Science.

The procrastination indicators and academic performance indicators were selected due to their broad nature, to allow for summative results after analysis. However, most articles used these indicators in line with their own needs, and adapted them accordingly. The meta-analysis prioritizes quantitative data at the expense of qualitative precision. Put differently, the result is statistically extremely robust, but compromises the conditions under

which the results are attained. A more stringent data collection process would introduce a greater variety of indicators, thereby reducing the certainty of the conclusions due to a smaller number of studies for each specific indicator.

To address these limitations, future research should implement the following strategies: employ multiple assessors for data selection and extraction to reach consensus and establish inter-rater reliability; incorporate additional tests for publication bias; expand database searches and integrate data from existing meta-analyses on the subject; establish more precise criteria for indicators to preserve their interpretive value; and include grey literature in the search parameters to ensure a comprehensive review.

The implications of this research are mainly related to academic settings, as the correlation is primarily focused on academic performance. The direct implications of this research are the following: procrastination among students should be monitored to identify students that could suffer in terms of academic performance; not all procrastination should be viewed as negative as active procrastination is positively related to academic performance; procrastination should be given more attention the older people get, as there seems to be a positive correlation between age and effect size.

The implications of this research enrich the existing body of knowledge regarding the interrelationship between procrastination and academic performance. It also provides an overview of the different procrastination indicators and their relationship with academic performance. The results of the subgroup analysis on the type of procrastination are also of interest for researchers in this field; researchers should account for different types of procrastination when using or developing an indicator for procrastination. The type of procrastination can greatly influence the correlation with academic performance.

Procrastination as a concept should be well defined, as the results of this research show that procrastination can refer to a variety of behaviours. These behaviours generally follow the line of delay of tasks, but with different intentions and surrounding behaviours that lead to different performance results. The example provided in this study relates to academic performance correlating differently with active and passive procrastination; however, future research may distinguish many more differences between types of procrastination.

The results of this study can also be used to theorise about the direct and indirect influences of procrastination on academic performance. The distinction between passive and active procrastination can serve as a comparison between procrastination associated with negative performance and procrastination associated with positive performance. Both forms of procrastination contain dilatory behaviour within their definition; however, active is regarded as more positive compared to passive. One can theorise that if both forms of procrastination delay tasks to the same extent, then the only difference between the two is the traits associated with them. If this statement is true, then one can assume that the indirect effects of procrastination determine the direction and strength of the relationship. However, further research needs to be conducted to verify that both forms of procrastination cause task delay to the same extent, and determine whether the definitions are related enough to compare.

## 5. Conclusions

The overall relationship between procrastination and academic performance is negative and small. The type of procrastination does matter when compared with academic performance, as active procrastination is positively related to academic performance, while passive procrastination is negatively related to academic performance. When measuring procrastination, one should consider that external measurements of procrastination correlate more with academic performance compared to participant-reported data, keeping in mind that these forms of data collection differ in procrastination facets. It can be inferred that the correlation between academic performance and procrastination is more pronounced among college students compared to those in secondary education.

**Author Contributions:** Conceptualization, N.S.K. and F.P.; methodology, N.S.K.; software, N.S.K.; validation, N.S.K., C.V.N. and F.P.; formal analysis, N.S.K.; investigation, N.S.K.; resources, N.S.K. and F.P.; data curation, N.S.K.; writing—original draft preparation, N.S.K.; writing—review and editing, N.S.K., C.V.N. and F.P; visualization, N.S.K.; supervision, F.P.; project administration, N.S.K.; funding acquisition, F.P. All authors have read and agreed to the published version of the manuscript.

**Funding:** This research received no external funding.

**Conflicts of Interest:** The authors declare no conflict of interest.

## Appendix A. A Summary of Measurement Tools Used to Assess Procrastination

**Table A1.** A comprehensive summary of procrastination indicators.

| Measurement | Abbreviation | Active or Passive Procrastination | How Procrastination Is Assessed | Test–Retest Reliability |
|---|---|---|---|---|
| Academic Delay Scale [58] | EDA | Passive, it reports the tendency to delay tasks considering the negative consequences. | Assesses delay tendency in academic tasks. | - |
| Academic Procrastination Scale [24] | APSM | Both, it has dimensions related to both types of procrastination. | Assess the type of procrastination using cognitive efficiency, peak experience, deliberate procrastination, preference for pressure, ability to meet deadlines, and outcome satisfaction to measure active procrastination and passive procrastination using fear of failure, taskaversiveness, and laziness. | - |
| Academic Procrastination [59] | APF | Passive, it reports on task avoidance, which is related to passive procrastination. | Assesses the avoidance of academic tasks using five self-reported items. | - |
| Academic Procrastination Questionnaire [60] | APQ | Passive, it relates to the delay of academic work without mentioning benefits. | Assesses procrastination of academic work and tasks using a questionnaire format. | - |
| Academic Procrastination Scale [61] | EPA | Passive, it relates to self-regulation failure and can therefore not be active. | Assesses academic procrastination through two dimensions named academic self-regulation and postponement of activities through a 12-item questionnaire. | - |
| Academic Procrastination Scale [62] | APSC | Passive, it aims to define procrastination in one dimension with a negative connotation. | Assesses academic procrastination behaviours using 19 self-assessment items to indicate greater or more procrastination behaviours. | $r = 0.89$ |
| Academic Procrastination Scale [9] | APSG | Passive, it assesses the negative aspects of academic procrastination. | Assesses student academic procrastination based on five self-assessment items. | - |
| Academic Procrastination Scale [63] | APSMT | Passive, it focuses on the difficulty to start an academic task. | Assesses academic procrastination with a focus on homework, examinations, and papers using seven items. | - |
| Academic Procrastination Scale [64] | APSO | Passive, it relates to the negative aspects of procrastination behaviour. | Assesses the probability of procrastination behaviour using 11 self-assessment items. | - |

Table A1. *Cont.*

| Measurement | Abbreviation | Active or Passive Procrastination | How Procrastination Is Assessed | Test–Retest Reliability |
|---|---|---|---|---|
| Academic Procrastination Scale-Short Form [65] | APSS | Passive, it notes the negative outcome of procrastination on academic tasks. | Assesses academic procrastination using five items on tendency to procrastinate on academic tasks. | - |
| Academic Procrastination State Inventory [66] | APSI | Passive, it has a focus on state procrastination with a negative connotation. | Assesses state procrastination using 13 items with facets such as delay, lack of energy, and concentration. | $r = 0.69$ [67] |
| Academic Time Management and Procrastination Measure [68] | ATMPM | Passive, it gives a rating to procrastination's relationship to poor time management. | Assesses behaviours related to time management (planning and monitoring) and procrastination using 14 self-report items. | |
| Active Procrastination Scale [30] | APSCM | Active, highlights the positive aspects of procrastination. The scale can be reverse keyed for passive procrastination. | Assesses four characteristics of procrastination items: satisfaction with an outcome, ability to meet deadlines, intention to procrastinate, and preference for time pressure using 16 self-assessment items. | $r = 0.80$ |
| Adult Inventory of Procrastination [69] | AIP | Passive, it highlights the negative relationship with time. | Assesses adult procrastination on the subscales: time loss, time management, and time commitment. | $r = 0.71$ [70] |
| Aitken Procrastination Inventory [71] | API | Passive, it is used to differentiate students from chronic academic procrastinators. | Assesses students and analyses their level of chronic academic procrastination using 21 items. | $r = 0.87$ [72] |
| Avoidance Reactions to a Deadline Scale [73] | ARDS | Passive, it focusses on avoidance instead of the positive aspect of delay. | Assesses avoidance reactions indicative of procrastination using 8 self-reported items. | - |
| Behavioural and Emotional Academic Procrastination Scale [74] | BEPS | Passive, it provides a rating based on negative aspects of delay and comfort. | Assesses the self-reported behavioural aspect of procrastination using three items on academic task delay and the emotional aspect of procrastination using three items on subjective discomfort. | $r = 0.62$ (delay) and $r = 0.52$ (subjective discomfort) |
| Conscientiousness Measurement [75] | CCM | Passive, it measures the opposite of procrastination behaviour. | Assesses self-reported conscientiousness using the facets industriousness, perfectionism, tidiness, procrastination refrainment, control, caution, task planning, and perseverance using 68 items. | - |
| Decisional Procrastination Scale [5] | DPS | Passive, negative aspects of decisional procrastination. | Assesses five items that relate to delay in coming to decisions and delay in implementing decisions. | $r = 0.69$ [76] |
| General Procrastination Scale [32] | GPS | Passive, keyed towards procrastination in everyday situations and tasks. | Assesses general procrastination using 20 items on everyday situations and tasks. | $r = 0.80$ [77] |
| Irrational Procrastination Scale [78] | IPS | Passive, as scores are lower when procrastination is not viewed negatively. | Assesses the self-reported feeling of experiencing an irrational delay. | $r = 0.84$ [79] |

**Table A1.** *Cont.*

| Measurement | Abbreviation | Active or Passive Procrastination | How Procrastination Is Assessed | Test–Retest Reliability |
|---|---|---|---|---|
| Melbourne Decision Making Questionnaire [80] | MDMQ | Passive, it is related to the negative results of procrastination. | Assesses decision-coping patterns using the factors procrastination, hyper-vigilance, buck-passing, and vigilance using 22 self-report items. | - |
| Metacognitive beliefs about Procrastination scale [26] | MaP | Both, relates to the positive and negative metacognitive beliefs about procrastination. | It assesses metacognitive beliefs about procrastination using 16 self-assessment items. | - |
| Multidimensional Academic Procrastination Scale [81] | MAPS-15 | Passive, it is related to the negative consequence of procrastination. | Assesses academic procrastination using the factors core procrastination, poor time management, and work disconnection. | $r = 0.85$ |
| Passive Procrastination Scale [23] | PPSC | Passive, it is meant to contrast with active procrastination. | Assesses academic procrastination using six self-reported items adapted from the DPS and APSI. | - |
| Procrastination Checklist Study tasks [28] | PCS | Passive, it is related to study behaviours and time in a negative fashion. | It is an external procrastination measurement using 12 study behaviours scored for both the intended and complete time. | - |
| Procrastination Assessment Scale for Students [6] | PASS | Passive, as student scores are lower when they do not think procrastination is a problem. | Assesses procrastination and to what extent it causes problems using 12 items and two subscales related to frequency and problem. | $r = 0.57$ |
| Procrastination Inventory [82] | PI | Passive, it denotes the negative connotation of procrastination. | Assesses procrastination using the four subscales controllability, expectation to change, motivation to change, and justification using 36 items. | - |
| Procrastination Log-Behaviour [83] | PLB | Passive, it has a focus on procrastination behaviour and satisfaction. | Assesses self-reported procrastination behaviour weekly with a satisfaction rating using 11 items. | - |
| Procrastination State in Academic Tasks [84] | PATS | Passive, it relates to negative aspects of procrastination behaviour and state of mind. | Assesses self-reported procrastination over the course of a week using the factors procrastination behaviour, fear of failure, and lack of academic motivation. | - |
| Procrastination Styles Questionnaire [85] | PSQ | Passive, it accounts for delay of academic tasks but not the benefit of delay. | Assesses behavioural procrastination using ten academic scenarios that are rated using the four responses: classic procrastination, non-academic productive procrastination, academic productive procrastination, and non-procrastination. | - |

**Table A1.** *Cont.*

| Measurement | Abbreviation | Active or Passive Procrastination | How Procrastination Is Assessed | Test–Retest Reliability |
|---|---|---|---|---|
| Pure Procrastination Scale [78] | PPS | Passive, it highlights the dysfunction of procrastination. | Assesses dysfunctional procrastination using 12 self-report items from different procrastination identifiers. | $r = 0.89$ [86] |
| Revised NEO Personality Inventory [87] | NEO-PI-R | Passive, it relates to the refrainment of negatively associated procrastination. | Assesses procrastination refrainment through items of the facet of self-discipline. | - |
| Student Learning Inventory [88] | SSLI | Passive, it highlights the negative aspect of procrastinative metacognition. | Assesses five self-assessment factors on metacognition named postdictive, predictive help, predictive no help, procrastinative, and piecemeal. | - |
| Test Procrastination Questionnaire [89] | TPQ | Passive, it focuses on the likelihood of procrastination over a test. | Assesses the likelihood of procrastination over a test using ten self-assessment items. | - |
| The Studying Procrastination Scale [90] | SPS | Passive, it views active procrastination as wrong by the definition of procrastination. | Assesses study procrastination of students and the self-assessed implications towards performance alongside affect and self-forgiveness. | - |
| The Unintentional Procrastination Scale [91] | UPS | Passive, it is related to unintentional procrastination and highlights the negative aspects of procrastination. | It assessed unintentional procrastination using 6 items. | - |
| Tuckman Procrastination Scale with 35 items [92] | TPS | Passive, no items related to positive aspects of procrastination. | Assesses 35 statements about the participant's relationship with procrastination including self-efficacy and self-regulated performance items. | $r = 0.90$ |

Note. This table gives a summary of most indicators of procrastination and how they assess procrastination, including the type of procrastination and the test–retest reliability. If the test–retest reliability was not found, then a hyphen is used instead. When a source is cited next to the test–retest coefficient, then the coefficient is extracted from a different source than the original article in the measurement section. The abbreviations have been gathered from the article in which the measurement tool was found, or the abbreviations have been composed from the first letters of each the measurement and when necessary the first letters of the author's name.

## Appendix B. An Overview of Included Studies

**Table A2.** Studies included in the meta-analysis.

| Number | Author | Year | Sample Size | Sample Age | Location |
|---|---|---|---|---|---|
| 1 | Alp and Sungur [93] | 2017 | 117 | College | Turkey |
| 2 | Artino et al. [94] | 2012 | 170 | College | United States |
| 3 | Atalayin et al. [95] | 2017 | 452 | College | Turkey |
| 4 | Balkis [96] | 2011 | 364 | College | Turkey |
| 5 | Balkis and Duru [97] | 2017 | 441 | College | Turkey |
| 6 | Balkis (a) [31] | 2013 | 290 | College | Turkey |
| 7 | Balkis (b) [98] | 2013 | 323 | College | Turkey |
| 8 | Balkis et al. [99] | 2012 | 281 | College | Turkey |
| 9 | Batool [100] | 2019 | 502 | College | Pakistan |
| 10 | Bolden and Fillauer [101] | 2020 | 114 | College | United States |
| 11 | Bong et al. [102] | 2014 | 304 | Secondary education | South Korea |
| 12 | Caratiquit and Caratiquit [103] | 2023 | 223 | Secondary education | Philippines |

**Table A2.** *Cont.*

| Number | Author | Year | Sample Size | Sample Age | Location |
|---|---|---|---|---|---|
| 13 | Cerezo et al. [104] | 2016 | 140 | College | Spain |
| 14 | Chen and Zeng [105] | 2022 | 566 | College | China |
| 15 | Chissmom et al. [106] | 1989 | 118 | College | United States |
| 16 | Chu and Choi [23] | 2005 | 230 | College | Canada |
| 17 | Clariana et al. [107] | 2012 | 171 | College | Spain |
| 18 | Clarke and MacCann [108] | 2016 | 457 | College | Australia |
| 19 | Corkin et al. [56] | 2011 | 206 | College | United States |
| 20 | Corkin et al. [109] | 2021 | 223 | College | United States |
| 21 | Cosnefroy et al. [110] | 2018 | 303 | College | France |
| 22 | Custer [111] | 2018 | 195 | College | United States |
| 23 | de la Fuente et al. [112] | 2021 | 430 | Secondary education | Colombia |
| 24 | Duru and Balkis [113] | 2017 | 348 | College | Turkey |
| 25 | Duru and Balkis [114] | 2014 | 261 | College | Turkey |
| 26 | Elias et al. [115] | 2005 | 145 | College | Malaysia |
| 27 | Estrada Araoz et al. [116] | 2020 | 47 | College | United States |
| 28 | Fernández Da Lama and Brenlla [117] | 2022 | 257 | College | Argentina |
| 29 | Franzen et al. [118] | 2021 | 6609 | Secondary education | Luxembourg |
| 30 | Gadosey et al. [119] | 2022 | 1556 | College | Germany |
| 31 | García-Ros et al. [9] | 2022 | 728 | Secondary education | Spain |
| 32 | Gareau et al. [120] | 2018 | 258 | College | Canada |
| 33 | Garzón-Umerenkova et al. [121] | 2018 | 363 | College | Spain |
| 34 | Ghayas et al. [122] | 2022 | 200 | College | Pakistan |
| 35 | Goroshit [123] | 2018 | 142 | College | Israel |
| 36 | Grunschel et al. [124] | 2016 | 635 | College | Germany |
| 37 | Han et al. [125] | 2023 | 1216 | College | United States |
| 38 | Hensley [126] | 2014 | 320 | College | United States |
| 39 | Hofer et al. [127] | 2012 | 697 | Secondary education | Germany |
| 40 | Job et al. [128] | 2015 | 145 | College | United States |
| 41 | Kandemir [129] | 2014 | 619 | College | Turkey |
| 42 | Karatas [130] | 2015 | 475 | College | Turkey |
| 43 | Kármen et al. [131] | 2015 | 162 | College | Romania |
| 44 | Kertechian [132] | 2018 | 404 | College | France |
| 45 | Kim and Nembhard [133] | 2019 | 59 | College | United States |
| 46 | Kim and Seo [134] | 2013 | 278 | College | South Korea |
| 47 | Kim et al. [26] | 2017 | 178 | College | Switzerland |
| 48 | Kindt et al. [135] | 2019 | 418 | Secondary education | Germany |
| 49 | Klassen et al. [136] | 2008 | 456 | College | Canada |
| 50 | Klingsieck et al. [137] | 2012 | 1396 | College | Germany |
| 51 | Kljajic and Gaudreau [138] | 2022 | 269 | College | Canada |
| 52 | Kljajic and Gaudreau [139] | 2018 | 208 | College | Canada |
| 53 | Kljajic et al. [140] | 2022 | 359 | College | Canada |
| 54 | Kljajic et al. [141] | 2017 | 510 | College | Canada |
| 55 | Kurtovic et al. [142] | 2019 | 227 | College | Croatia |
| 56 | Lim [143] | 2016 | 214 | College | United States |
| 57 | Lubbers et al. [144] | 2010 | 9811 | Secondary education | Netherlands |
| 58 | MacCann et al. [75] | 2009 | 275 | College | United States |
| 59 | Macher et al. [145] | 2011 | 147 | College | Austria |
| 60 | Martín-Antón et al. [146] | 2022 | 794 | College | Spain |
| 61 | Martinie et al. [147] | 2022 | 236 | College | France |
| 62 | Michinov et al. [148] | 2011 | 83 | Adults | France |
| 63 | Moon and Illingworth [149] | 2005 | 303 | College | United States |
| 64 | Moon et al. [150] | 2020 | 96 | College | United States |
| 65 | Morris and Fritz [151] | 2015 | 67 | College | United Kingdom |
| 66 | Paechter et al. [152] | 2017 | 225 | College | Austria |
| 67 | Pekpazar et al. [153] | 2021 | 378 | College | Turkey |
| 68 | Pilotti et al. [154] | 2022 | 609 | College | Saudi Arabia |
| 69 | Purwanto and Natalya [155] | 2019 | 239 | College | Indonesia |
| 70 | Ragusa et al. [156] | 2023 | 991 | Secondary education | Spain |

**Table A2.** *Cont.*

| Number | Author | Year | Sample Size | Sample Age | Location |
|---|---|---|---|---|---|
| 71 | Rikoon et al. [87] | 2016 | 426 | College | United States |
| 72 | Roig and DeTommaso [157] | 1995 | 58 | College | United States |
| 73 | Sæle et al. [158] | 2016 | 379 | College | Norway |
| 74 | Sage et al. [159] | 2021 | 96 | College | United States |
| 75 | Saman and Wirawan [160] | 2021 | 1670 | College | Indonesia |
| 76 | Seo [a] [18] | 2011 | 172 [a] | College | South Korea |
| 77 | Seo [a] [52] | 2012 | 172 [a] | College | South Korea |
| 78 | Shaked and Altarac [161] | 2022 | 145 | College | Israel |
| 79 | Steel et al. [162] | 2001 | 152 | College | United States |
| 80 | Suárez-Perdomo et al. [163] | 2022 | 1784 | College | Spain |
| 81 | Sun and Kim [164] | 2022 | 157 | College | United States |
| 82 | Tian et al. [165] | 2020 | 108 | Secondary education | China |
| 83 | Tian et al. [166] | 2021 | 3511 | College | China |
| 84 | Tice and Baumeister [167] | 1997 | 104 | College | United States |
| 85 | Tisocco and Liporace [168] | 2022 | 928 | College | Argentina |
| 86 | Wesley [169] | 1994 | 244 | College | United States |
| 87 | Westgate et al. [85] | 2016 | 1104 | College | United States |
| 88 | Wolters [170] | 2004 | 525 | Secondary education | United States |
| 89 | Wolters and Hussain [171] | 2014 | 213 | College | United States |
| 90 | Wu [172] | 2020 | 78 | College | Taiwan |
| 91 | Xu [173] | 2023 | 1072 | Secondary education | China |
| 92 | Yang et al. [174] | 2020 | 242 | College | Estonia |
| 93 | You [175] | 2015 | 569 | College | South Korea |
| 94 | Yu et al. [176] | 2021 | 465 | Secondary education | China |
| 95 | Zhang and Zhang [177] | 2022 | 55 | College | China |
| 96 | Zhang et al. [178] | 2022 | 265 | College | United States |

Note. The sample size of each article is reported as the highest number of participants reported per correlation coefficient in cases of pooled participants and the combined number of participants in cases of multiple separate studies to report the number of unique participants per article. [a] These studies share sample sizes but are analysed using different indicators.

**Appendix C. Funnel Plot**

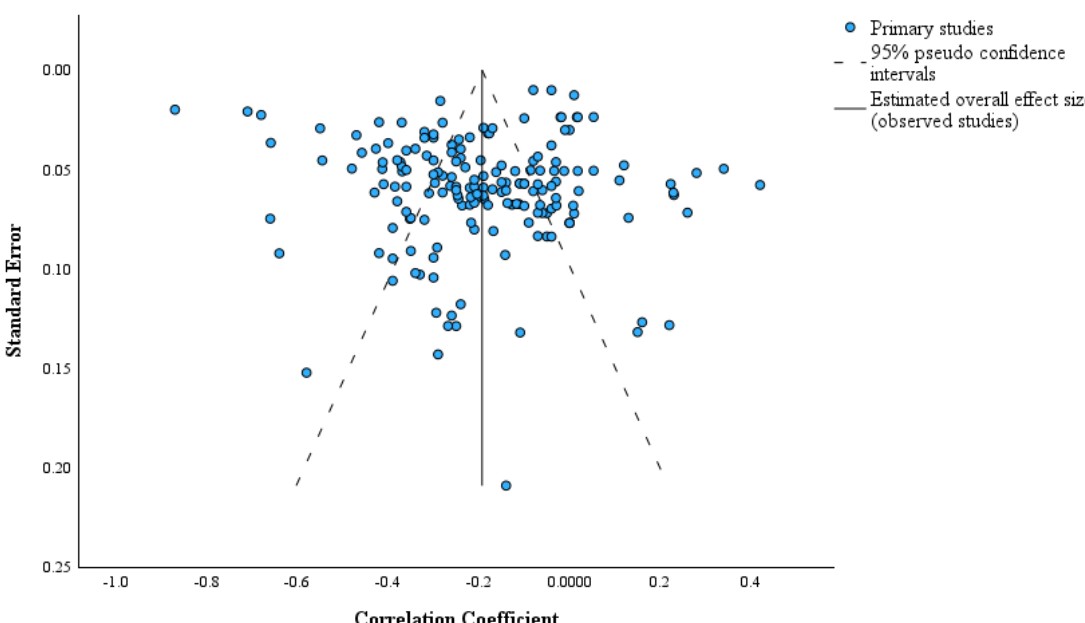

**Figure A1.** The funnel plot related to the Egger's regression test for correlation coefficient and standard error. Note. The blue dots in the funnel graph represent individual correlation coefficients.

## Appendix D. The Results of Subgroup Statistical Tests

**Table A3.** Results of effect size (*r*) estimates for subgroup academic performance indicators.

| Academic Performance Indicator | *k* | Effect Size (*r*) | Std. Error | Z-Value | Sig. (2-Tailed) | 95% Confidence Interval Lower | Upper |
|---|---|---|---|---|---|---|---|
| ACT | 2 | 0.04 | 0.0700 | 0.575 | 0.565 | −0.097 | 0.177 |
| Average Grade | 23 | −0.27 | 0.0292 | −9.388 | <0.001 | −0.332 | −0.217 |
| Course grade | 25 | −0.28 | 0.0583 | −4.831 | <0.001 | −0.396 | −0.168 |
| Coursework | 1 | −0.39 | 0.1060 | −3.680 | <0.001 | −0.598 | −0.182 |
| Exam | 39 | −0.16 | 0.0293 | −5.566 | <0.001 | −0.220 | −0.106 |
| Flesch-Kincaid Grade Level | 1 | −0.27 | 0.1287 | −2.082 | 0.037 | −0.520 | −0.016 |
| General Average | 1 | −0.19 | 0.0650 | −2.891 | 0.004 | −0.315 | −0.061 |
| GPA | 60 | −0.18 | 0.0219 | −8.284 | <0.001 | −0.224 | −0.138 |
| Homework assignment | 7 | −0.14 | 0.0919 | −1.476 | 0.140 | −0.316 | 0.044 |
| Research specific | 9 | −0.12 | 0.0559 | −2.084 | 0.037 | −0.226 | −0.007 |
| SATs | 5 | 0.01 | 0.0124 | 0.977 | 0.329 | −0.012 | 0.037 |
| Written report | 3 | −0.33 | 0.0666 | −4.967 | <0.001 | −0.461 | −0.200 |
| Overall | 176 | −0.19 | 0.0153 | −12.569 | <0.001 | −0.223 | −0.163 |

SATs = Standard Assessment Tests.

**Table A4.** Results of effect size (r) estimates for subgroup procrastination indicators.

| Procrastination Indicator | *k* | Effect Size (*r*) | Std. Error | Z-Value | Sig. (2-Tailed) | 95% Confidence Interval Lower | Upper |
|---|---|---|---|---|---|---|---|
| Absence | 2 | −0.63 | 0.0800 | −7.893 | <0.001 | −0.788 | −0.475 |
| APF | 2 | −0.18 | 0.1400 | −1.289 | 0.197 | −0.455 | 0.094 |
| API | 13 | −0.27 | 0.0393 | −6.759 | <0.001 | −0.343 | −0.189 |
| APSCM | 22 | 0.07 | 0.0356 | 2.050 | 0.040 | 0.003 | 0.143 |
| APSG | 1 | −0.24 | 0.0349 | −6.986 | <0.001 | −0.312 | −0.176 |
| APSMT | 1 | −0.21 | 0.0802 | −2.618 | 0.009 | −0.367 | −0.053 |
| APSO | 2 | −0.25 | 0.0543 | −4.673 | <0.001 | −0.360 | −0.147 |
| APSS | 7 | −0.30 | 0.0275 | −10.917 | <0.001 | −0.355 | −0.247 |
| CCM | 2 | −0.08 | 0.0998 | −0.835 | 0.404 | −0.279 | 0.112 |
| Days between assignments | 1 | −0.25 | 0.0647 | −3.786 | <0.001 | −0.372 | −0.118 |
| Days to complete | 1 | −0.39 | 0.0586 | −6.566 | <0.001 | −0.500 | −0.270 |
| Days to start | 1 | −0.13 | 0.0677 | −1.875 | 0.061 | −0.260 | 0.006 |
| Days-hand-in | 1 | −0.35 | 0.0748 | −4.703 | <0.001 | −0.499 | −0.205 |
| Dilatory behaviour | 10 | −0.24 | 0.0296 | −8.085 | <0.001 | −0.297 | −0.181 |
| DPS | 1 | −0.20 | 0.0637 | −3.139 | 0.002 | −0.325 | −0.075 |
| EDA | 1 | −0.38 | 0.0660 | −5.757 | <0.001 | −0.509 | −0.251 |
| EPA | 1 | −0.30 | 0.0324 | −9.272 | <0.001 | −0.363 | −0.237 |
| GPS | 11 | −0.31 | 0.0614 | −5.059 | <0.001 | −0.431 | −0.190 |
| Inactive time | 2 | −0.61 | 0.0567 | −10.679 | <0.001 | −0.716 | −0.494 |
| IPS | 9 | −0.21 | 0.0404 | −5.172 | <0.001 | −0.288 | −0.130 |
| Late submission | 2 | −0.58 | 0.1050 | −5.489 | <0.001 | −0.782 | −0.371 |
| NEO-PI-R | 1 | 0.12 | 0.0479 | 2.504 | 0.012 | 0.026 | 0.214 |
| PASS | 21 | −0.14 | 0.0381 | −3.746 | <0.001 | −0.217 | −0.068 |
| PI | 1 | −0.04 | 0.0838 | −0.477 | 0.633 | −0.204 | 0.124 |
| PPS | 4 | −0.27 | 0.0707 | −3.845 | <0.001 | −0.410 | −0.133 |
| PPSC | 2 | −0.37 | 0.0691 | −5.322 | <0.001 | −0.503 | −0.232 |
| PSQ | 4 | −0.09 | 0.0507 | −1.828 | 0.067 | −0.192 | 0.007 |
| Research specific | 25 | −0.18 | 0.0417 | −4.375 | <0.001 | −0.264 | −0.101 |
| SPS | 1 | −0.35 | 0.0744 | −4.702 | <0.001 | −0.496 | −0.204 |
| SSLI | 2 | −0.04 | 0.2050 | −0.217 | 0.829 | −0.446 | 0.357 |
| Time pressure reactivity | 2 | −0.21 | 0.0922 | −2.259 | 0.024 | −0.389 | −0.028 |
| TPS | 20 | −0.24 | 0.0247 | −9.646 | <0.001 | −0.286 | −0.190 |
| Overall | 176 | −0.19 | 0.0153 | −12.569 | <0.001 | −0.223 | −0.163 |

Note. The abbreviations found in the 'Procrastination Indicator' row are explained in Appendix A.

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
