# Peer review of "The Influence of Active and Passive Procrastination on Academic Performance: A Meta-Analysis"

_education, doi:10.3390/educsci14030323_

Round 1
Reviewer 1 Report
Comments and Suggestions for Authors
1. It is not clear from the abstract the rationality of the present study: many studies for 20 years indicated small or no relationship while recent studies indicated something different. Which is the role of a meta analysis? This is obvious at the following sections of the manuscript (mainly at the Discussion), however it would be useful to be revealed at the abstract, as well.
2. Authors do not use the reference style which is used by the MDPI.
3. It is useful the initial definition of the academic procrastination
4. It is interesting to have the initial research on the domain and the recent ones.
5. The presentation of the issue and the main concepts at the first pages lead to the focus of the present study and underline its significance.
6. The method which was used was appropriate and we have a well structured presentation. It is interesting that we have data for a long period (1975-2023).
7. excellent presentation of the method of meta analysis
8. it is not clear for me why the 885 articles were characterized as irrelevant to the current research.
9.It is extremely "clear" and based on "logical steps" the presentation of the discussion. It is not usual to have so "helpful" for the reader presentation which leads fluently and with arguments at the answering of the posed research questions.
10. Similarly authors show how the present limitations can be the indications for the following up studies.
Reviewer 2 Report
Comments and Suggestions for Authors
This is a very interesting paper documenting a meta-analysis conducted to establish the effect of active procrastination on academic performance. There were times where I needed to check the dictionary for the definitions of some of the words used but this means I have increased my vocabulary, and I don’t think they need any further elaboration by the authors. I have indicated where I think a further explanation is needed as it is important to the narrative in your paper. You have given a very detailed analysis of the method you used and included the range of statistical analysis you used to gather your data.
In the results you appear to rely on the reader accessing the large appendix to find information. The appendix does not appear to be formatted correctly and it was really difficult to find any information that you referred to in your results section. Unfortunately, this decreases the readability of your paper. A lot of information in the appendix could be summarized and included in tables in the main text. Consider summarizing the important information or choose key points and place in tables in the appropriate positions within the text. The results section needs to be rewritten so that the relevant information is more easily accessed by the reader.
In the reference section you have included papers used in the meta-analysis but were not cited in the paper. You need to check whether this is normal practice.
Below you will find some typos listed and some suggestions to improve the readability of this paper.
Line 42 – should it be ‘as’ rather than ‘at’?
Line 82 – Could you please add some more information or details about the term ‘neuroticism’? This term is important in the statements about correlation with both passive and active procrastination.
Line 103 – there needs to be a space between’ tasks” and the bracket
Lines 200 – 230
· Figure 1 has moved to the left of the page on the pdf copy. The text in the boxes appears to be larger than the normal text and double spaced and thus takes up lots of space. Is there a reason why this is not the same sized (or smaller) font and single spaced?
· I initially noticed that the list in the box in Figure 1 [Reports excluded (n = 125)] appeared to be different to the list given under the Method section (lines 175 – 181). However, when I read the Note (lines 232 – 242) I realized that you are referring to criteria.
· I assume the criteria you are referring to are those given lines 175 – 181. The link between the reasons and the criteria needs to be made much clearer. Maybe state Criteria 1, Criteria 2 etc in lines 175 – 181 and then refer back to them when talking about the Reasons in Figure 1.
· Why is the note section not included in normal text as part of the method section? Consider including a Table that has a list of the reasons in one column and the matching criteria in the other.
Line 320 – I think you mean average not ‘averaged’.
Line 323 – I’m not sure why you have placed the other data related to the sample size in Appendix C. You have omitted the word ‘in’. As a reader I was frustrated when trying to hunt through your huge appendix to find what you were referring to when you said there was other data. Why aren’t the key points about the sample size summarized in a Table placed here in the document?
Line 325 – If it is important for the reader to see the positive trend please include figure 1 in the text not the appendix.
Lines 351 – 366 are difficult to read as there is so much statistical information in this paragraph. It appears to replicate what is given in Table 1.
Line 367 – Table 1 does not fit the page and is difficult to unpack. If the font was smaller it would be almost impossible to read. Consider the important information that the reader needs to make sense of your paper and remove the other columns.
Line 379 – again the reader should not have to access the summary information for themselves from Appendix A. If it is important place the information in the text.
Line 419 – I think you mean the second research question.
Line 487 – remove extra space between ‘their’ and ‘broad’.
Line 489 – Not sure what you mean by the term ‘qualitative precision’. Could you please elaborate?
Lines 543 – 599
· This is a very large Appendix. It is not formatted so that a reader can easily access the information referred to in the paper. You do not need to double spaced text in tables.
· Are all the Appendices needed? Why?
· Some of these Appendices could be summarized and included in the main text. Consider which columns include important information for the reader and delete the rest.
Lines 601 – 1073
· You have included papers that you used in the meta-analysis but have not cited in the paper. You need to check whether this is normal practice.
Comments on the Quality of English LanguageJust a couple of typos which are highlighted in general feedback.
